# Corporate Social Responsibility and Entrepreneurial Ventures: A Conceptual Framework and Research Agenda

Régis Y. Chenavaz [1,*] , Alexandra Couston [2], Stéphanie Heichelbech [3], Isabelle Pignatel [1] and Stanko Dimitrov [4]

[1] Finance, Accounting and Economics Department, Kedge Business School, 13288 Marseille, France; isabelle.pignatel@kedgebs.com
[2] Marketing Department, Kedge Business School, 33405 Talence, France; alexandra.couston@kedgebs.com
[3] Management Department, Université Paris Cité, LIRAES, 75014 Paris, France; stephanie.bugaut@etu.u-paris.fr
[4] Department of Management Sciences, University of Waterloo, Waterloo, ON N2L 3G1, Canada; sdimitrov@uwaterloo.ca
* Correspondence: regis.chenavaz@kedgebs.com

**Abstract:** Corporate social responsibility (CSR) and entrepreneurship are two essential topics in the current business landscape. However, despite the growing literature on these topics, there needs to be more comprehensive understanding of how they are related. In this conceptual article, we explore the linkages between CSR and entrepreneurship. First, we provide a definition and scope of entrepreneurship and then discuss the literature on CSR, highlighting different ways that businesses can engage in CSR. We argue that CSR and entrepreneurship are closely related, and propose a conceptual framework to understand how CSR can be integrated into the entrepreneurial process. Additionally, we identify three key areas of research in this emerging field: (1) the motivations for entrepreneurs to engage in CSR; (2) the impact of CSR on entrepreneurial ventures; and (3) the role of CSR in social entrepreneurship. We conclude with a discussion of our conceptual framework's theoretical and practical implications, as well as future research directions for scholars and practitioners interested in CSR and Entrepreneurship.

**Keywords:** entrepreneurship; corporate social responsibility; survey; triple bottom line; sustainable development

## 1. Introduction

Entrepreneurship and corporate social responsibility (CSR) have become two increasingly essential concepts, which have been studied in various fields. Before continuing, we build on the literature to define each concept. Entrepreneurship is the activity of designing, launching, and running a new business or enterprise, especially one that is small and started with limited resources [1,2]. CSR is business' commitment to improving the quality of life for their employees, their families, and the local community [3]. More broadly, CSR has essential effects on environmental issues, poverty eradication, employment creation and labor practices, environmental protection, education, and human development [4,5]. Building on the above definitions, we can see how these two concepts are related [3]. One of the critical aspects of entrepreneurship is innovation, which means that entrepreneurs are always looking for new ways to carry projects out and new products or services to offer. This spirit of invention is also what drives CSR. Businesses that are committed to CSR are always looking for new ways to make a positive impact on the world. Innovation is one of many things that these two concepts have in common. Another common aspect is that entrepreneurship and CSR are about more than just making money [6]; they are both about making a difference. Entrepreneurs start businesses because they want to make a difference in the world, creating something new and making a positive impact. CSR is also about making a difference, as indicated by its definition.

Various studies and fields have identified the connection between entrepreneurship and CSR. In the business field, for example, CSR can be seen as a by-product of entrepreneurial activities [7]. Entrepreneurs start companies to make a profit; however, to do so, they must satisfy a variety of stakeholders, including employees, customers, suppliers, and the local community. This way, CSR appears as a natural consequence of the entrepreneurial process. Economics has also studied the relationship between entrepreneurship and CSR. Previous research in [8] argues that CSR can be a competitive advantage for firms, as it can help them differentiate themselves from their rivals. This advantage exists because consumers are increasingly interested in purchasing from socially responsible firms. Therefore, firms that engage in CSR may gain a competitive advantage over those that do not.

Sociology has also explored the link between entrepreneurship and CSR. Sociologists have long argued that developing new businesses can help revitalize communities and create new jobs [9]. This revitalization is because entrepreneurship can lead to new products and services that benefit society. In addition, entrepreneurship can also help create new employment opportunities, which can positively impact social cohesion. Education has also looked at the connection between entrepreneurship and CSR. For example, several studies have examined how entrepreneurship education can help develop students' social and environmental awareness [10,11]. These studies suggest that entrepreneurship education can play a role in teaching young people about the importance of social and ecological responsibility. Clearly, there is a connection established between entrepreneurship and CSR in different fields. This connection is essential because it suggests that CSR is not just a burden for businesses but can also be a source of competitive advantage.

John Elkington coined the phrase "people, planet, and profit" in 1994 to describe the triple bottom line (TBL) and sustainability goals. TBL theory proposes that there should be three (people, planet, and profit) rather than one bottom line. Today, organizations understand success is reflected in more than just their financial statements. The TBL looks at three Ps—people, planet, and profit—to assess an organization's impact [6,12]. People: How do we treat our employees, customers, and other stakeholders? Planet: What is our environmental footprint? Profit: Is our company profitable and sustainable? The three Ps are interconnected [13]. What affects one affects the others. For instance, an organization polluting the environment would not only damage the planet but could face increased costs and regulatory pressure, affecting its bottom line. On the flip side, an organization that practices sustainable measures can help the planet and benefit from improved efficiency and reduced costs. TBL is more than a theory; it is a way of conducting business that is being adopted by organizations across the globe.

The triple bottom line is a framework for sustainable business practices that consider the impact of a company's activities on people, the planet, and profits [6,11,12]. The "people" aspect refers to the social dimension of sustainability, which includes considering the well-being of employees, customers, communities, and other stakeholders affected by the business. The "planet" aspect concerns the environmental impact of a company's operations, including issues such as carbon emissions, waste management, and resource conservation. Finally, the "profit" aspect recognizes the economic dimension of sustainability, which entails making a profit while also balancing the social and environmental impacts of the business. By addressing all three aspects of the triple bottom line, companies can create long-term value for themselves and society. Our conceptual article is based on the existing links between these three themes (people, planet, profit) through the lens of entrepreneurship and CSR. The term CSR has been widely used in recent times [3,14]. It is often confused with "sustainability" or TBL; however, there is a noteworthy distinction between them. While sustainability focuses on an organization's environmental stewardship and accountability, CSR goes further, covering all three aspects of the triple bottom line: people, planet, and profit [15]. CSR, in its most basic form, is about how businesses and entrepreneurs can perform in a way that not only generates financial returns but also contributes to society and the environment. Although this idea is not novel, it has been

receiving more attention lately as businesses are increasingly held accountable for their effect on society and nature.

Scholars recognize that entrepreneurship is a complex concept to define [1,16]. Ref. [17] proposes that entrepreneurs are unique individuals, capable of identifying and addressing inefficiencies in the economic system. To put it broadly, entrepreneurs can detect minor discrepancies in market operations and be the first to benefit from them. In practice, the entrepreneur can take opportunities, outperform others, and develop new production, marketing, and organization methods. CSR is important for businesses and entrepreneurs for various reasons [18]: Primarily, it is simply the correct thing to do. Businesses and entrepreneurs must act morally and responsibly [19]. This entails transparency about their operations, reducing their environmental footprint, and treating their employees equitably. CSR is beneficial for business. Customers, investors, and employees are increasingly interested in working with and backing firms that are socially and ecologically conscious [20]. By incorporating CSR into their business model, businesses and entrepreneurs can access this increasing market and enhance their earnings [21]. Finally, CSR is essential for the future of our planet.

Businesses and entrepreneurs also have an obligation to address global issues such as climate change, poverty, and inequality [22]. By improving the conditions in which they operate, they can positively contribute to the world and shape a better future for all. CSR is a growing field, encompassing various fields such as business, sustainability, and law [23]. This is due to an increasing awareness of the need to take responsibility for the environment and society [24]. CSR is the notion that companies should consider the social and environmental impacts of their activities and make efforts to minimize any adverse effects [25]. This includes reducing emissions and ensuring ethical sourcing [26]. There is evidence that CSR can be beneficial to businesses [2]. It can lead to improved reputation, increased profits, and improved employee morale [12,23].

CSR can help businesses build trust with stakeholders, which is essential for lasting success [27,28]. While the rewards of CSR are evident, there remains much to learn regarding the most effective ways of implementing and developing CSR initiatives [20]. Recent literature can help fill that gap by examining CSR programs and policies to identify best practices and suggesting how businesses can most effectively participate in CSR. Away from academia, the Business and Sustainable Development Commission has published a comprehensive report on the business case for CSR, and the International Business Leaders Forum has released a series of information on the issue. Moreover, several journals focus on CSR research, such as *Sustainability*, the *Journal of Business Ethics*, and the *Business & Society Review*.

The aim of this article is to present a novel conceptual framework centering around the three classical pillars of CSR: people, planet, and profit [13,18,20,22,25]. As far as we know, this standard CSR structure has never been applied to entrepreneurship, making our article an innovative addition to the existing literature. Furthermore, our article defines sustainable entrepreneurship as a business that can satisfy the present needs without jeopardizing the ability of future generations to meet their own needs. Finally, we assert that sustainable entrepreneurship is not only crucial for the survival of our planet but also beneficial for the economy and society as a whole.

We contribute to the literature on entrepreneurship and CSR by providing a conceptual article along the three classical pillars of CSR. More precisely, Section 2 presents entrepreneurship and people, Section 3 links entrepreneurship and the planet, and Section 4 explores on entrepreneurship and profit. Eventually, Section 5 discusses the research implications, directions and conclusions.

## 2. Entrepreneurship and People

Social entrepreneurship encompasses a wide range of activities in developed and developing countries aimed at addressing deficiencies in public policies, social issues ignored by the private sector, and economic hardships that affect many people. Innovative

and visionary individuals with human and technical skills employ hybrid approaches that bridge the public and private sectors to make up for institutional shortcomings. Table 1 presents selected articles on the relationship between entrepreneurship and people.

In order to better understand the concept of social entrepreneurship and to identify its drivers, academic literature provides complementary definitions. Ref. [29] conducted a semantic study of social entrepreneurship, social entrepreneurs, and social enterprise and found that these terms are interchangeable in English. Ref. [30] defines social entrepreneurship as being focused on people and based on economic efficiency and social purpose. Refs. [31–34] suggest that social entrepreneurship has emerged as a response to new societal challenges and a means to generate social value.

Since the 1980s, there has been a growing focus on socially oriented enterprises in policy making but less so in research [35]. This gap may be due to sustainable entrepreneurship creating new employment opportunities [36], providing easy access to new services, promoting integration, and supporting local economies. Ref. [2] has identified a lack of research on the social and environmental context of sustainable entrepreneurship, with only a few studies on the subject [37–42]. These studies mainly focus on organizations that pursue the dual bottom line of economic and social/ecological goals, with [37] concentrating on the potential exploitation of such firms.

Social entrepreneurs play a crucial role in shaping public policies. At a global level, social entrepreneurs can shape public policies via their activities and opportunities. At an organizational level, they can fill gaps in provision from public and private organizations, improving the lives of individuals. At an individual level, the success of social enterprises depends on the traits and abilities of the social entrepreneur. In the remainder of this section, we explore each of these levels in turn.

**Table 1.** Selected articles for entrepreneurship and people.

| Authors | Contributions | Advancement of New Knowledge, a Guide to Practice and Further Research |
|---|---|---|
| [29] | Semantic study of social entrepreneurship, social entrepreneurs, and social enterprise. | Clarification of the terms used to define social entrepreneurship. |
| [30] | Definition of social entrepreneurship: to be focused on people and based on economic efficiency and social purpose. | A human-centric approach to social entrepreneurship. Development of long-term economic benefits for communities. |
| [31–34] | Social entrepreneurship as a response to new societal challenges and a means to generate social value. | Social entrepreneurship as a response to new societal challenges and an effective way to generate social value. Social entrepreneurship as a tool to address specific societal challenges. |
| [2] | Identification a lack of research on the social and environmental context of sustainable entrepreneurship. | Identifying the need for more research on sustainable entrepreneurship and its social and environmental context. |
| [37] | Potential exploitation of firms that pursue the dual bottom line of economic and social/ecological goals. | Exploration of ways to ensure that businesses pursuing the dual bottom line are not exploited for financial gain. |
| [43] | Examination of the differences between social business models and entrepreneurship models, as their economic goals may vary. | Critical analysis of the evolution of entrepreneurship and business concepts. |
| [44] | How social enterprises integrate with capitalism without challenging it. | Demonstration of the coexistence of innovation, social commitment, and profit in social entrepreneurship. Investigation about how social enterprises can integrate with capitalism without compromising their social and environmental goals. |

**Table 1.** *Cont.*

| Authors | Contributions | Advancement of New Knowledge, a Guide to Practice and Further Research |
|---|---|---|
| [45] | Examination of how non-profit organizations use commercial or managerial expertise to generate income with creative strategies. | How non-profit organizations (1) aim to create social value instead of personal gain or remuneration for shareholders, (2) develop innovative approaches to imitate the activities of for-profit companies. |
| [46] | Importance of a process-oriented approach to achieving the three goals of sustainable entrepreneurship. | Significance of targeting social and economic objectives in social entrepreneurship. |
| [47] | Universal definition of social entrepreneurship and outline of a social business model. | Demonstration about how social entrepreneurs navigate between the economic, social, and ecological dimensions. |

*2.1. Global Level*

Social entrepreneurship is a response to the limitations of liberal logic and the need to balance economic and social components in a globalized world facing various crises. It promotes unity and has gained prominence in developed nations to address escalating social and environmental issues [48]. While it may have a limited effect on business operations, it targets specific social and economic challenges [49] and environmental issues [50]. The potential of social entrepreneurship in times of economic hardship lies in its ability to bridge the gap between the social and market economies and to form partnerships with local authorities to overcome challenges. Establishing regional hubs and collaborations can lead to complementary relationships between various economic sectors [51].

However, social entrepreneurship must also question its ability to sustain itself and remain a force in the public and political sphere [52]. Though it has gained recognition as a socio-economic paradigm, its success may come from a favorable context for private initiatives and support from various actors. The economic crisis of 2008 led to its use as a means of combating unemployment and as an instrument for reconfiguring public policies. Practical social entrepreneurship requires "champions" who can identify strategies that will enhance the lives of individuals or communities [53]. These champions can either generate original solutions or replicate successful projects. Social entrepreneurs take the initiative in areas where public authorities need to address needs, and their initiatives may coincide with incomplete public service objectives. Examples of social entrepreneurship include hospices, healthcare, aid for disadvantaged young people, rehabilitation, support for families and the unemployed, and access to housing. Some projects are created in response to a crisis, while others emerge through close contact with disadvantaged communities. They may be local, national, or international efforts.

Social entrepreneurship is the identification of initiatives that can create social value and address social issues, as recognized by various authors [31,42,54–59]. It may apply in the private, for-profit sector, commercial entities with a social aim, non-profit organizations, and hybrid structures linking profit and non-profit objectives. It is an essential concept for addressing social issues that transcend the internal concerns of businesses and may originate from macro- and micro-environments.

To propose a legal definition, in France, the law No. 2014-856 of July 31, 2014 (JORF, 2014) indicates that [60]: "The social and solidarity economy is a mode of entrepreneurship and economic development adapted to all areas of human activity to which members are legal persons governed by private law who meet the conditions following accumulations:

1.    A goal pursued other than the sole sharing of profits;
2.    Democratic governance, defined and organized by the articles of association, providing for information and participation, the expression of which is not only linked to

their capital contribution or the amount of their financial contribution, of partners, employees, and from stakeholders to achievements, of the company;

3. Management following the following principles:

    (a)    The profits are mainly devoted to the objective of maintaining or business development;

    (b)    The obligatory reserves constituted, indivisible, cannot be distributed. "

Entrepreneurship driven by adversity is often referred to as "necessity entrepreneurship", which is frequently overlooked in the literature. Yet, it plays a crucial role in providing employment and satisfying basic needs [61]. The creation of quality jobs and wealth generation is linked to the importance of human capital and theories of human resource management, entrepreneurship, and leadership, which together form the concept of "Humane Entrepreneurship" (HumEnt). HumEnt is in line with the United Nations' Sustainable Development Goals (SDGs) [62]. Ref. [56] highlights that social entrepreneurship is a process that transforms or catalyzes opportunity from the innovative combination of resources needed to address unmet or inadequately met social needs. Ref. [63] argues that social exclusion is a complex issue, and entrepreneurship can only partially address it. Although a positive link exists between business creation, ownership, and economic growth (as highlighted in the Global Entrepreneurship Monitor Report), self-employment does not guarantee a certain level of income or social advancement. Ref. [64] discusses a U.S. Department of Labor-funded project to increase self-employment and small business creation for disabled people in Onondaga County, New York.

Social entrepreneurship is also a marker of development in countries with prevalent poverty and inequality. Ref. [65] uses Morocco's National Initiative for Human Development (INDH) to illustrate that social entrepreneurs are primarily motivated by economic and financial gain as it provides them with the means to meet their basic needs. Female entrepreneurship can significantly impact social welfare in developing countries, such as through poverty reduction, birth control, children's and particularly girls' education, gender relations, and household management. However, there is also a darker side to entrepreneurship, particularly for women entrepreneurs in developing countries, who often live in precarious conditions and have limited protection [66]. Ref. [67] examines the impact of public funding on social integration businesses and hybrid and entrepreneurial organizations, which take on a public service role in Switzerland. These organizations face competition, economic risks, administrative standardization, and enforcing contracts, which can detract from solidarity economy principles and social responses to social needs in an inclusive manner [68]. Several contextualizations of entrepreneurship exist; for instance, gender [69], minority status [70], immigration status [71], and disability have been largely overlooked [72]. We will touch on some of these points next.

Social entrepreneurship can play a crucial role in addressing the challenges faced by people with disabilities and minority groups. However, it is crucial to recognize that these individuals' obstacles and limitations may differ from those faced by the general population. Research has shown that people with disabilities are less likely to succeed in launching a business due to their existing obstacles and limited opportunities [73]. These obstacles include a need for more financial resources, social networks, and educational backgrounds compared to their peers. Despite the incentive structures of public policies, the challenges for disabled entrepreneurs remain [74]. Similarly, minority business owners face unique challenges in starting and growing their businesses. Historically, their reach did not exceed specific tasks, and their customers were typically from the same demographic [70]. However, with the availability of higher education, members of minority communities have been able to engage in various activities and overcome some of these limitations.

In rural and disadvantaged regions, social entrepreneurship can also empower women and expand their autonomy [75,76]. The Mahaul project, for example, has demonstrated the potential of emancipatory social entrepreneurship to improve the economic, social, and human standing of women in disadvantaged regions by freeing them from existing social and cultural norms [76].

Overall, it is crucial to recognize that social entrepreneurship can be a powerful tool for addressing the unique challenges faced by people with disabilities and minority groups. However, it is crucial to understand these individuals' specific obstacles and limitations and tailor strategies accordingly.

### 2.2. Organizational Level

Researchers such as the authors of [38] investigate the contrast between opportunities for generating and innovating in social entrepreneurship, which depend on factors such as a social mission [77], institutional or social barriers to entering a specific social market, and the entrepreneur's past knowledge [41]. These factors can increase an individual's knowledge of a social issue, leading to entrepreneurial opportunities. Additionally, the knowledge useful to create entrepreneurial opportunities comes from different individuals. It may increase through sharing and collective action [41,78].

Resolving a social problem is the driving force behind establishing a social enterprise, which can bring together the resources essential to its success [30]. Realizing the three objectives of sustainable entrepreneurship—economic, social, and ecological—can be a source of ambiguity, arbitration, and conflicts [79]. Social entrepreneurs have the ambition and ability to gather the necessary human and material resources to make their projects a reality [53]. These entrepreneurs may have selfless motivations and come from diverse backgrounds, such as religious practitioners, civilians, retirees, or those active in social welfare and community causes [80].

A conceptual framework highlighting the need to manage social entrepreneurship at the individual and organizational/inter-organizational levels is proposed in [81]. This approach stresses the importance of management similar to commercial enterprises, even though the creation of social value rather than profit. Funding for this activity may be provided by subsidies, patrons, entrepreneurs, and networks of actors involved in solidarity and social economy, which can sometimes be structured as more complex organizations such as cooperatives or mutuals.

Better communication about social entrepreneurship and supportive and instructional programs could lead to more individuals having the necessary capabilities for the success of social entrepreneurs' projects [53]. Social entrepreneurship can take various forms, such as a non-profit business with a commercial aim, a for-profit entity with a social or organizational mission, or a hybrid organization combining entrepreneurial and social practices [82]. Social entrepreneurship projects may comprise teams from diverse backgrounds, from the public or private sector [83]. However, the project leader who starts the initiative, gathers resources and involves a community may need the profile to manage the project over the long term. Successful social enterprises can be deployed through spin-offs, thanks to managers.

Sustainable entrepreneurship, which has economic, social, and ecological goals of equal significance, is examined in [2]. Other authors focus on entrepreneurship, the entrepreneurial process, and how opportunities and individuals are linked, along with these three goals [46,84–88]. Social entrepreneurship has economic and social goals as its primary concern, with the distinction being the number of purposes, the type of organizations, and the notion of equity [89]. Despite these differences, commonalities may appear as the perception of an ecological or social issue or opportunity and the visionary's ability to develop a solution or business model to address this need or opportunity.

Personal funds or the support of family members are often the primary sources of funding for sustainable entrepreneurship ventures [37]. However, they also note that alternative forms of financing, such as crowdfunding or public funding, have become more prevalent. These companies must consider the additional cost of implementing sustainable practices, which may limit their target market. Ref. [43] explores the evolution of the concepts of entrepreneurship and business and examines the differences between social business models and entrepreneurship models, as their economic goals may vary.

Traditional buying criteria such as performance, functionality, and design in marketing may align with social or environmental objectives. They may also incorporate ecological

or social elements to become a part of a sustainable approach to products or services [90]. According to [44], a social enterprise integrates with capitalism without challenges. The three main driving forces of social entrepreneurship (innovation, social commitment, and profit) are blended with personal and societal conditions of the social and solidarity economy to be examined collectively. This idea demonstrates that these three facets coexist, innovation is a motivator, and that profit may be a motivator, as is the case with for-profit companies [91].

The skills and commercial or managerial expertise used by non-profit organizations in a market economy appear to generate income with creative strategies [58,92]. The primary aim is to create social value instead of personal gain or remuneration for shareholders, which motivates social entrepreneurship (e.g., [45]). This aim includes innovation or the introduction of new approaches to imitate the activities of for-profit companies.

The importance of a process-oriented approach to achieving the three goals of sustainable entrepreneurship is stressed in [46]. This process starts with awareness and recognition of opportunities, followed by creating and implementing new products, services, or prices to bring societal, economic, and ecological benefits. These opportunities for change are clear in realizing the flaws (inadequacy of the offer, business inefficiencies, asymmetry of information) that exist in the markets, which are a source of social and ecological issues and, therefore, also show many opportunities. Ref. [93] further underscores the importance of the process viewpoint in social entrepreneurship, which should target social and economic objectives.

An in-depth examination of social entrepreneurship by developing a universal definition for the concept, gaining worldwide popularity, is given in [47]. The authors define it as "a way of accessing market resources to fulfill a social need" and outline their social business model as "a for-profit, non-profit, private or public enterprise, established under a particular legal structure, aiming to meet a social need (social outcome) through creating market resources (economic outcome), using managerial techniques and exploiting social innovations". Through their actions, social entrepreneurs navigate between the economic, social, and ecological dimensions, which can be sources of tension and identity conflicts. They also lead them to develop strategies for managing these tensions and conflicts to ensure sustainability [94].

According to [95], a project's success in entrepreneurship is measured by its implementation and development, which involves transforming an idea into desired outcomes. This transformation requires three key elements: conveying a vision of an ideal future state in an unpredictable setting, establishing a purpose, direction, and objectives to attain this objective by mobilizing resources, and ensuring that the aim is achieved by appealing to the parties essential to success. For the social entrepreneur, the ability to persuade, enlist, revive, and share the vision of a more promising future for a community is crucial for success.

Social entrepreneurship can promote the growth of social capital by encouraging a more inclusive and diverse community, including those with physical or social disabilities [96]. According to [97], social capital is primarily based on people's capacity to work together to achieve a common goal, collectively or through institutions. For Leadbeater, social entrepreneurship is a response to the inadequacies of the welfare state in addressing a range of social issues. Creative communities can aid in generating social capital from different and diverse groups, including those with physical or social disabilities. Dedicated social entrepreneurs who can bring together a range of assets to meet the community's needs without a profit motive lead these projects. They aim to create a project that provides value to the most disadvantaged in a given area.

Some authors emphasize that the collective aspect based on connections and alliances is essential [98]. Ref. [99] studied female entrepreneurs' use of social media. Their literature review highlights the significant number of studies on these practices, mainly in developing countries, intending to start a business. However, networking activities appear as a vital

strategic management tool [100] for acceptance and integration at the cultural, political, and social levels.

### 2.3. Individual Level: Profile and Training

Social entrepreneurship is a field that combines the aims of economic growth and social impact. Ref. [30] defined it as a pioneering, socially valuable venture that can take place in the non-profit, business, or government sectors. They distinguish social entrepreneurship from traditional business entrepreneurship by highlighting its unique context, individuals and resources, and clients or social beneficiaries. The mission of social entrepreneurship closely relates to social goals and performance measurement tools, and the motivations of those involved are not solely focused on monetary gain or profit. Social beneficiaries often have limited purchasing power, which requires the offer to be crafted and communicated differently.

Leadership is an essential quality of a social entrepreneur, according to [53]. They argue that social entrepreneurs must possess vigor, dedication, and the ability to bring people together around their vision and shared desirable objectives. They may face obstacles such as limited time to devote to the venture, difficulty building alliances around their vision, lack of technical and managerial expertise in project implementation, and lack of training in project organization and advancement.

Entrepreneurs and social entrepreneurs share many similarities, including ambition, leadership, problem-solving capabilities, disruptive thinking, and persuasive communication skills [96]. Furthermore, [77,101] emphasize that the drive and unselfishness of the individual entrepreneur are essential for successful social entrepreneurship. Nevertheless, "Motivation and social purpose are not enough for the social enterprise. Without strategic and managerial skills, the objective of creating and sustaining jobs underlying cannot be implemented" [102].

The argument that many minority entrepreneurs merge personal aspirations and public betterment in running their businesses appears in [103]. They blend profit ambitions with social objectives. Ref. [104] suggests that self-employment and entrepreneurship can be a form of vocational rehabilitation for the disabled and those recovering from illness. People with mental disabilities or who lack the intellectual aptitude expected by society can evolve through entrepreneurship, expressing themselves, honing their abilities, and cultivating their character.

Those with disabilities are more likely to become entrepreneurs than those without disabilities [74]. Entrepreneurship provides an opportunity that may not be available through salaried jobs. Ref. [105] highlighted that entrepreneurship could be a vital tool for improving living conditions in poverty areas and can be associated with personal satisfaction and pleasure.

The capabilities required to become a sustainable entrepreneur are vast and diverse, ranging from innate abilities such as physical, social, and organizational skills [106] to those acquired through experience or education [107–109]. To develop these skills, comprehensive and in-depth training is essential [110], typically offered in business schools for entrepreneurship education and environmental education departments for sustainability education.

To be an effective sustainable entrepreneur, one must possess the ability to combine entrepreneurship and sustainability [111,112]. These skills should allow individuals to approach real-world problems, identify opportunities, and address relevant environmental issues [113–116].

The competencies needed for sustainability-focused entrepreneurship have been extensively studied and discussed by [115], who identified seven fundamental abilities: system thinking, foresight, normative, embracing diversity and interdisciplinarity, interpersonal, action, and strategic management. Subsequent studies [113,117–119] have further expanded on these competencies and their application in the realm of sustainability.

In recent years, sustainability research has increasingly focused on the abilities of entrepreneurs and aspiring entrepreneurs [110]. With regard to educational opportunities and training for those interested in pursuing a career in social entrepreneurship, [35] noted a limited number of programs, such as the well-regarded and comprehensive entrepreneurship or social economy programs offered by Stanford and Harvard Universities in the USA, as well as postgraduate courses in these fields at Cambridge, Oxford, and Southampton in the UK.

Ref. [120] explores the social competencies crucial for the success of social and solidarity economy (SSE) entrepreneurs who have completed a master's diploma program. They stress the importance of self-awareness, creativity, and establishing and maintaining relationships with employees. Pursuing a course of study in the SSE field before training provides credibility and mentoring, which can enhance self-confidence and proficiency in management strategies [121].

According to [122], sustainable entrepreneurial competency that aligns with the Sustainable Development Goals (SDGs) is a vital factor influencing entrepreneurial intentions and the formation of aspirations to consider the SDGs, particularly for students in higher education.

The description of the mission of the social entrepreneurship chair at the prestigious French business school ESSEC is provided in [123]. At the school, students motivated by ideas of value, seeking significance, and environmental and social values find a curriculum that promotes the management of cities and territories to foster interaction and human connection. However, an underlying aspect of the SSE sector is related to influence or politics.

Ref. [124] examines the agile teaching formats developed by a Bavarian university since 2013. The program aims to encourage entrepreneurship and promote the acquisition of critical skills through new teaching formats and project-based learning methods.

The 2030 Agenda for Sustainable Development defines 17 goals with 169 economic, social, and environmental targets to ensure human rights. Universities, particularly business and management education, and businesses are essential drivers for achieving these Sustainable Development Goals (SDGs). Thus, universities must mobilize their managers, professors, and students, as implementing the SDGs is only possible through coordinated and integrated participation [125].

To conclude this section, sustainable entrepreneurship is an increasing and significant area that could help the world by creating jobs, producing revenue, and improving the environment. This field intersects with topics such as global economic growth, human resources management, and sustainability at international, organizational, and individual levels. From an individual-only point of view, sustainable entrepreneurship is about establishing a business model that is socially and environmentally responsible. At an international level, it means organizations must create and execute policies and practices that support and promote sustainable entrepreneurship. At an organizational level, organizations must establish and implement policies and procedures that encourage employee participation in sustainable entrepreneurship. Lastly, from a global standpoint, sustainable entrepreneurship necessitates a joint effort between individuals and organizations to create a sustainable future. The field of sustainable entrepreneurship is proliferating and there is still much to be uncovered about how to build and sustain socially and environmentally responsible businesses.

### 2.4. Research Gaps

Although social entrepreneurship has gained recognition for its potential to address social and economic challenges, several research gaps still need further exploration. Firstly, there is a need for more research on the impact of social entrepreneurship at a global level. While it has been acknowledged that social entrepreneurs can shape public policies and create opportunities, their influence's scope and impact require further understanding.

Secondly, sustainability is a significant concern in social entrepreneurship, and more research is required to assess the long-term sustainability of social entrepreneurship initiatives. Although social entrepreneurship has gained recognition as a socio-economic paradigm, its ability to sustain itself and remain relevant in the public and political sphere needs further questioning. Therefore, there is a need for more research to explore the sustainability of social entrepreneurship initiatives in different contexts and environments.

Thirdly, there is a need to explore the potential of social entrepreneurship in addressing the challenges faced by people with disabilities, minority groups, and other disadvantaged communities. Research has shown that these groups face several obstacles and limitations that hinder their business success. Therefore, more research is needed to explore how social entrepreneurship can be used to address the challenges faced by these groups.

Fourthly, there is a need to examine the different contextualizations of entrepreneurship, including gender, minority status, immigration status, and disability. Although some studies have touched on these points, there remains a significant gap in our understanding of how these factors influence social entrepreneurship. Future research could explore the unique challenges and opportunities faced by individuals from diverse backgrounds in social entrepreneurship and identify ways to address these challenges.

## 3. Entrepreneurship and Planet

Entrepreneurship and the planet have been linked since the beginning of economic development. Entrepreneurs have been responsible for creating new products, services, and business models that have improved human welfare; at the same time, however, they have also been responsible for environmental degradation. The Brundtland Report, published in 1987 (https://sustainabledevelopment.un.org/content/documents/5987our-common-future.pdf) (accessed on 25 May 2023), marked the starting point of attention being paid to ecological dilemmas and introduced the concept of sustainable development. In recent years, the one-pillar model of sustainable development, which prioritizes the ecological dimension, has been gaining momentum. This momentum is because natural resources and ecosystems are finite, and the economy is the leading cause of environmental problems. Table 2 shows selected articles on the relationship between entrepreneurship and planet.

The one-pillar model of sustainable development strongly emphasizes the planet, as it prioritizes the ecological dimension [126]. It relies on the idea that economic development must balance the protection of natural resources and ecosystems. This approach highlights the need to balance economic growth with social and environmental considerations.

This section will focus on how entrepreneurship can contribute to sustainable development by protecting the planet. We will examine sustainable entrepreneurship and sustainable performance and explore how entrepreneurship may protect the planet. We will also discuss the benefits and further actions of entrepreneurship. Entrepreneurship plays a critical role in addressing environmental challenges and achieving sustainable development. Entrepreneurs are responsible for creating new products, services, and business models that can improve human welfare while protecting the planet.

### 3.1. Sustainable Entrepreneurship and Sustainable Performance

Sustainable entrepreneurship and sustainable performance are two key concepts that have gained significant attention in the literature on entrepreneurship and the planet. We explore here the different definitions of sustainable entrepreneurship, how the literature defines environmental performance, and the theoretical background for sustainable entrepreneurship.

**Table 2.** Selected articles for entrepreneurship and planet.

| Authors | Contributions | Advancement of New Knowledge, Guide to Practice and Further Research |
|---|---|---|
| [127] | Identifying four different 'entrepreneurial cultures' among EU countries, underlining that culture is one of the key elements explaining differences in context conditions for starting a business venture. | Understanding the cultural context for entrepreneurship and identifying differences in context conditions for starting a business venture. |
| [128] | Green entrepreneurship refers to activities that consciously address environmental and social problems through the implementation of entrepreneurial ideas, amidst high risks and with the expectation of a net positive impact on the environment and financial sustainability. | Identifying and promoting green entrepreneurship as a means of addressing environmental and social problems through entrepreneurial ideas. |
| [129] | Green ventures have a more positive economic and social impact and are less harmful or even beneficial to environmental quality when compared to conventional new ventures. | Encouraging the creation of green ventures as a means of achieving a net positive impact on the environment and financial sustainability. |
| [130] | Tools such as toxic corporate releases, socially responsible indices, and Environmental, Social and Governance-based ratings can help individuals and organizations track and improve their environmental impact. | Using environmental performance measurement tools to identify areas for improvement and set targets for reducing environmental impact. |
| [131] | Many individuals are choosing to become green entrepreneurs, who consciously address environmental or social problems and achieve a net positive impact on the environment and financial sustainability. | Encouraging individuals to become green entrepreneurs as a means of addressing environmental or social problems and achieving a net positive impact on the environment and financial sustainability. |
| [90] | Introduction of the concept of sustainable development and emphasized the need to balance economic growth with social and environmental considerations. | The starting point of attention paid to environmental dilemmas. The need to explore ways to achieve sustainable development that balances economic, social, and environmental considerations. |
| [126] | Introduction the one-pillar model of sustainable development, which prioritizes the ecological dimension. | The need to balance economic growth with social and environmental considerations, with an emphasis on protecting natural resources and ecosystems. The role of entrepreneurs in addressing environmental challenges and achieving sustainable development by creating new products, services, and business models that improve human welfare. The need to explore how entrepreneurs can contribute to sustainable development, particularly in protecting the planet. |
| [84,87,132,133] | Different definitions for sustainable entrepreneurship, including environmental entrepreneurship, green entrepreneurship, and sustainable entrepreneurship. | The agreement that green entrepreneurship should be defined in terms of the adopted technological line of production or a firm's activities. Green entrepreneurship as a storytelling process through which an entrepreneur obtains support from stakeholders to pursue their goals. |

The literature on entrepreneurship and the planet provides several definitions for sustainable entrepreneurship, including ecopreneurship [132,134] environmental entrepreneurship/environpreneurship [84], green entrepreneurship [133], and sustainable entrepreneurship [37,135,136].

Despite the different definitions, there is agreement that green entrepreneurship should be defined in terms of the adopted technological line of production or a firm's activities [137]. Green entrepreneurship should also be a storytelling process through which an entrepreneur obtains support from stakeholders to pursue their relationship [114]. However, green entrepreneurship has no universally accepted definition [138].

The literature on sustainable entrepreneurship also addresses the concept of environmental performance. However, there has yet to be an agreement on a possible definition of environmental performance. Scholars still need to agree on a definition of this phenomenon [126]. Even though there is no definition, there have been attempts to measure environmental performance. Environmental performance is aggregated in a composite indicator (Social Progress Index) and mixed with social performance [139]. Ref. [140] confirms that there are at least two dimensions in CEP (corporate environmental performance): environmental managerial and environmental organizational practices.

The literature on sustainable entrepreneurship is divided into two main streams [126]:

- Environmentally oriented literature: This stream focuses on entrepreneurs' attitudes towards their enterprises' environmental goals and policies. These entrepreneurs follow their motivation to earn financial benefits by helping to decrease environmental problems and ecological degradation [84,134,136,141]. In this stream, the planet is a secondary purpose [37].
- Sustainability-oriented literature: This stream explores the relationship between sustainable development and entrepreneurship. Sustainable entrepreneurs seek to solve societal and environmental problems through their entrepreneurial activities [37,85,88,126]. The planet is the primary purpose of this stream [84,128].

Ecologically sustainable entrepreneurship is described [126] as "the process of identifying, evaluating and seizing entrepreneurial opportunities that minimize a venture's impact on the natural environment and therefore create benefits for society as a whole and local community". Gast's approach considers only one pillar, that is, the environment.

The question of how ecosystems can specifically promote sustainable entrepreneurship and contribute to the Sustainable Development Goals (SDGs) set by the United Nations is a neglected issue [127]. The authors of [131] regret that the environmental performance concept is reduced to the ecological impacts caused by companies. However, there is a positive relationship between energy efficiency and entrepreneurship in countries exhibiting high density in these two variables—e.g., Australia and the United Kingdom [129].

Sustainable entrepreneurship and sustainable performance are complex concepts that have attracted significant attention in the literature on entrepreneurship and the planet. While there are different definitions for sustainable entrepreneurship, the literature generally agrees that it should be defined in terms of the adopted technological line of production or a firm's activities and should be a storytelling process through which an entrepreneur obtains support from stakeholders to pursue their goals [137]. Additionally, the literature hosts two main streams: environmentally oriented and sustainability oriented. The former focuses on entrepreneurs who follow their motivation to earn financial benefits by helping to decrease environmental problems and ecological degradation. In contrast, the latter focuses on entrepreneurs who seek to solve societal and environmental issues through their entrepreneurial activities.

On the other hand, measuring environmental performance is a more challenging task. The literature is still young and fragmented, and there needs to be an agreement on a definition. However, several attempts have tried to measure environmental performance through composite indicators such as the Social Progress Index [139] and by including dimensions such as product and process improvement, relationship with stakeholders, legal conformity, and its financial impacts.

Sustainable entrepreneurship and performance are crucial topics that need further exploration and understanding. The literature suggests that sustainable entrepreneurship should be defined in terms of the adopted technological line of production or a firm's activities and should prioritize the planet as a primary goal [137]. Additionally,

measuring environmental performance is a complex task requiring more research and definition agreement.

### 3.2. How May Entrepreneurship Protect the Planet?

Entrepreneurship can play a crucial role in protecting the planet by encouraging the development of green products and practices and promoting good living for flora and fauna. The trajectory of green entrepreneurship (GE) sends a strong signal to others in the business community, raising awareness of the need to green their business processes. This awareness can generate more environmentally friendly products and practices and, ultimately, a more sustainable future.

At the global level, there are variations in the level of entrepreneurial activity between countries. Societies' orientation towards entrepreneurship differs; some have higher entrepreneurial activity rates than others. Ref. [130] examined common patterns among EU countries. They found at least four different 'entrepreneurial cultures', underlining that culture is one of the key elements explaining the differences in context conditions for starting a business venture.

At the organizational level, green entrepreneurship may mean activities that consciously address environmental and social issues through the implementation of entrepreneurial ideas, with the expectation of a net positive impact on both the environment and financial sustainability [142]. Studies have shown that green ventures have a more positive economic and social impact and are less harmful or even beneficial to environmental quality when compared to conventional new experiences [143].

Entrepreneurship at the individual level can also play a crucial role in protecting the planet. One way is via the measurement of environmental performance. Tools such as toxic corporate releases, the ratio of recycled waste to total toxic waste generated, and socially responsible indices or environmental, social and governance-based ratings can help individuals and organizations track and improve their environmental impact [144].

These tools can help identify areas where improvements are possible and to set targets for reducing environmental impact. For example, the KLD Research and Analytics, Trucost, and Sustainable Asset Management (SAM) ratings are commonly used by academics to measure environmental performance. SAM ratings are helpful in benchmark countries and regions, while Trucost provides data for organizations such as governmental agencies and influential companies [145].

In addition, many individuals are choosing to become green entrepreneurs. Green entrepreneurship relates to activities that consciously address environmental or social problems and achieve a net positive impact on the environment and financial sustainability [142].

Research suggests that compared to conventional new ventures, new green ventures have a more positive economic and social impact and are less harmful or even beneficial to environmental quality [143]. This element highlights the potential for individuals to make a real difference in protecting the planet through entrepreneurial activities.

Global, organizational, and individual entrepreneurship can play a crucial role in protecting the planet. By promoting green business practices, recognizing sustainable opportunities, and measuring and improving environmental performance, entrepreneurs can help to create a more sustainable future. Green entrepreneurship has a strong potential to contribute positively to the environment, economy, and society. Governments, organizations, and individuals need to support and encourage this form of entrepreneurship as it will lead to a more sustainable future for all.

### 3.3. Benefits of Green Entrepreneurship and Further Necessary Actions

Green entrepreneurship offers numerous benefits for organizations, both in terms of efficiency within the firm and in terms of building community goodwill. One key benefit is the opportunity to improve efficiency within the firm by implementing environmentally friendly practices and processes. This benefit may save costs, increase productivity, and offer a more sustainable business model overall [129].

Another benefit of green entrepreneurship is the creation of community goodwill. Organizations can establish positive relationships with their host communities, employees, and other stakeholders by addressing environmental and social issues [142]. This improved relationship can lead to increased support and loyalty from these groups, which can help build a more prosperous and sustainable business [114].

In addition, green entrepreneurship also allows for a consistent open feedback loop, where organizations can receive and disseminate helpful information from and to the general public [142]. This feedback can help organizations stay informed about their stakeholders' needs and concerns and can allow them to make more informed decisions about their business practices.

To further support the development of green entrepreneurship, it is vital to create a knowledge database for effectively disseminating current green information and clarifying conceptual entrepreneurship definitions [142]. This knowledge will help to ensure that organizations have access to the information and resources they need to implement environmentally friendly practices and processes. Additionally, it will promote a shared understanding of what green entrepreneurship means, which can, in turn, help to encourage more widespread adoption of these practices. This next step will help to ensure that organizations have access to the information and resources they need to implement environmentally friendly practices and processes.

In conclusion, sustainable entrepreneurship is necessary for preserving the planet and promoting a shared understanding of green entrepreneurship. The one-pillar model of sustainable development prioritizes the ecological dimension and is gaining momentum and attracting more pillars due to the finite nature of natural resources and ecosystems. Green entrepreneurship is an essential element of sustainable development, as it encourages the development of green products and practices to protect the planet while creating financial benefits. By implementing environmentally friendly procedures and processes, organizations can establish positive relationships with their host communities, employees, and other stakeholders while improving efficiency within the firm. Ultimately, green entrepreneurship allows individuals to make a real difference in protecting the planet and promoting good living for flora and fauna, driving innovation and ingenuity, benefiting the planet.

*3.4. Research Gaps*

The literature on sustainable entrepreneurship and sustainable performance has made significant progress, but critical gaps still require further exploration. Firstly, there needs to be more consensus on the definition of green entrepreneurship, and scholars need to agree on a definition that goes beyond the technological line of production or a firm's activities. Similarly, while several attempts have been made to measure environmental performance through composite indicators, a universally accepted definition still needs to be developed.

Secondly, research is needed to understand how ecosystems can specifically promote sustainable entrepreneurship and contribute to achieving the United Nations' Sustainable Development Goals (SDGs). Additionally, there is a need to recognize the positive relationship between energy efficiency and entrepreneurship in countries that exhibit high density in these two variables, which is often overlooked when focusing solely on environmental impacts caused by companies.

Thirdly, more research is needed to identify the benefits of sustainable entrepreneurship and the actions that can be taken to promote it. Although entrepreneurs play a crucial role in addressing environmental challenges and achieving sustainable development, further exploration is needed to identify the full range of benefits and to develop strategies to promote sustainable entrepreneurship. By addressing these gaps, future research can contribute to advancing the field of sustainable entrepreneurship and enhancing our understanding of its potential to drive sustainable development.

In conclusion, sustainable entrepreneurship and performance are complex concepts that require further exploration and understanding. While there have been several attempts

to define and measure these concepts, more research is needed to identify the benefits of sustainable entrepreneurship and the actions that can be taken to promote it. Additionally, there is a need for more research on how ecosystems can specifically promote sustainable entrepreneurship and contribute to the SDGs.

## 4. Entrepreneurship and Profit

Entrepreneurship and profit have been widely studied in the literature, with the focus being on the financial performance of firms. According to [146], pure economic profit for a firm is the difference between outputs and inputs. While profit is a crucial aspect of entrepreneurship, it is not always the primary goal. In sustainable entrepreneurship, profit seems to be a constraint or a strategic tool for achieving sustainability within the firm [147,148]. The literature supports this perspective on corporate sustainability, which has focused on understanding how sustainable development policies affect firms' competitive advantage or disadvantage and profits [149]. Table 3 depicts selected articles on the relationship between entrepreneurship and profit.

**Table 3.** Selected articles for entrepreneurship and profit.

| Authors | Contributions | Advancement of New Knowledge, Guide to Practice and Further Research |
|---|---|---|
| [149] | Traditional view: profit maximization as the sole objective of firms and their workers, employees, managers, and entrepreneurs. | Consider alternative perspectives on profit to balance sustainability and profitability. |
| [150] | The concepts of "economic profit" and "entrepreneur" are interconnected. | Identify the role of entrepreneurship in profit-making. |
| [151] | Entrepreneurs are motivated by profit and strive to anticipate future needs, identify better techniques, innovate, and take risks. | Analyze how entrepreneurs balance profitability and sustainability. |
| [16] | Summary of different views of profit in neoclassical, ownership, Austrian, and entrepreneurship theory. CSR is possible in entrepreneurial firm theory but optional. | Consider CSR an optional strategy in entrepreneurship. |
| [144] | Sustainable entrepreneurship requires balancing the sometimes-conflicting goals of generating a profit and promoting sustainability. Entrepreneurs can achieve this balance by prioritizing innovation and business strategies. | Prioritize innovation and business strategies to balance profitability and sustainability. |
| [88] | Examined social entrepreneurs' motivations to understand profit's role in their decision making. | Analyze the role of profit in social entrepreneurship. |
| [152] | Compared the median cash incomes of entrepreneurs and employees in Australia and found that entrepreneurs have lower base incomes and continue to earn less than employees throughout their careers. | Analyze the non-monetary benefits of entrepreneurship. |
| [153] | Entrepreneurs are increasingly incorporating CSR practices and strategies into their companies. | Consider the impact of CSR practices on sustainability. |
| [154] | Explored the impact of new businesses on sustainability in the context of family firms. | Analyze the impact of new businesses on sustainability. |

Sustainable entrepreneurship offers opportunities for entrepreneurs to generate profit through environmentally friendly business models. These models aim to provide goods or services that positively impact the planet, such as developing products made from recycled materials or providing services that help reduce greenhouse gas emissions. Additionally, sustainable entrepreneurs can make money by implementing strategies that reduce the negative environmental impact of their business activities [150]. These strategies include

identifying opportunities to reduce waste, conserve resources, and minimize pollution, and then developing and implementing plans to take advantage of these opportunities.

One way to make money with sustainable projects is to develop and sell products or services that help businesses reduce their environmental impact. These products and services include energy-efficient products, waste reduction and recycling solutions, and consulting services to assist businesses in implementing sustainable practices. Another way to make money with sustainable projects is by investing in businesses working to develop and implement sustainable practices. By investing in these types of businesses, entrepreneurs can support and contribute to the growth of a sustainable economy.

Eventually, entrepreneurship and profit can coexist in a sustainable context. Sustainable entrepreneurs can make money by developing environmentally friendly business models, implementing strategies to reduce their environmental impact, and investing in sustainable businesses.

### 4.1. What Is Profit for the Sustainable Entrepreneur?

The concept of profit has been a central topic in economic and business literature for decades. According to the traditional view, profit maximization is the sole objective of firms and their workers, employees, managers, and entrepreneurs. This notion is at the heart of any discussion on the firm's role in society [151], and the economic literature broadly covers the origin and measurement of this concept. For example, [152] posits that the concepts of "economic profit" and "entrepreneur" are interconnected, and [155] argues that entrepreneurs are motivated by profit and strive to anticipate future needs, identify better techniques, innovate, and take risks.

However, the perspective on profit in sustainable entrepreneurship differs from this traditional view. For sustainable entrepreneurs, profit is not the ultimate goal but a constraint or a strategic tool for achieving sustainability within the firm [148]. Ref. [17] summarizes the different views of profit in the context of neoclassical, ownership, Austrian, and entrepreneurship theory, highlighting the perspective of profit from the entrepreneurial point of view. In this perspective, ownership and control belong to shareholders or a corporation, management is professional, the social mission is to produce the goods and services demanded, and the goal of the entrepreneurial project is to make positive profits and increase them over time. The key to success in entrepreneurship is awareness, creativity, and innovation. Vranceanu's work highlights that CSR is possible in entrepreneurial firm theory but optional.

In short, profit represents only a goal among others for sustainable entrepreneurs but a constraint or strategic tool for achieving sustainability within the firm. This perspective differs from the traditional view that sees profit as the goal of firms and entrepreneurs.

### 4.2. Is Profit a Goal or a Constraint for the Sustainable Entrepreneur?

The concept of entrepreneurship has traditionally been associated with the pursuit of profit and economic and financial returns, as described by [155]. However, in recent years, the idea of sustainable entrepreneurship has emerged, which combines the concepts of sustainability and entrepreneurship. Ref. [147] emphasizes that this type of entrepreneurship requires balancing the sometimes-conflicting goals of generating a profit and promoting sustainability.

Research has shown that entrepreneurship could be more attractive in terms of wages and material and financial rewards [16]. Scholars have also debated whether social entrepreneurs should aim to make a profit. Ref. [91] examined social entrepreneurs' motivations to understand profit's role in their decision-making.

A study [156] compared the median cash incomes of entrepreneurs and employees in Australia and found that entrepreneurs have lower base incomes and continue to earn less than employees throughout their careers. It was also reported that entrepreneurs earn 35% less than employees. Non-material and non-financial benefits of entrepreneurship have

also been studied by [157–159], who found that entrepreneurs have higher job satisfaction compared to employees.

Profit is often considered the primary goal of companies. However, French firms, as highlighted in [160,161], aim to develop new business lines or new products that are socially innovative or legitimize their place in the community.

Sustainable entrepreneurship, as demonstrated by [162] and reinforced by [163], is not solely related to companies' financial results. This fact suggests that sustainable entrepreneurship is more than just profit-oriented. Sustainability is becoming increasingly important for companies and is now part of the development of business plans [153].

Entrepreneurial status provides a sense of well-being that is not linked to material income but to a sense of autonomy, self-expression, and creativity [16]. Entrepreneurship represents a means of achieving an independent way of life.

Entrepreneurs may address environmental or social issues to generate profit [46,84]. By creating environmentally sustainable or socially responsible solutions, entrepreneurs can generate profit while addressing the harmful effects of their business practices.

Sustainable entrepreneurship requires balancing profit and sustainability [147]. They propose that entrepreneurs can achieve this balance by prioritizing innovation and business strategies. By focusing on these two areas, entrepreneurs can increase their firms' profitability while developing altruistic ideas or actions to address new market opportunities.

For opportunistic entrepreneurs, the primary goal is to build a business that generates profit [154]. In this context, sustainability opens new markets and increases profitability.

To meet the expectations of shareholders and other stakeholders, entrepreneurs are increasingly incorporating CSR practices and strategies into their companies. Ref. [164] notes the limited research on the impact of new businesses on sustainability. Ref. [165] explores this topic in the context of family firms. Since the late 2000s, consumers and legal bodies have increasingly used CSR practices to hold companies accountable for their actions [166]. Ref. [167] reviews the literature on sustainability in family firms, which are often entrepreneurial structures, and suggests that these firms may use sustainability as a means of advertising or to protect their reputation [168].

In a nutshell, entrepreneurship is traditionally associated with the pursuit of profit. However, sustainable entrepreneurship has emerged as a new way of thinking in which profitability and sustainability goals are balanced. Research has shown that entrepreneurship is only sometimes financially rewarding but is often associated with non-material benefits such as job satisfaction.

*4.3. The Genuine Goals of the Sustainable Entrepreneur*

The conventional perception is that the objective of entrepreneurship is to achieve financial gain [155]. However, entrepreneurs may also pursue non-financial goals such as autonomy, the pursuit of personal ideas, job satisfaction, and responsibility. These non-financial goals play a significant role in understanding the complexity of entrepreneurship.

Research on social entrepreneurship has primarily focused on understanding the relationship between social entrepreneurship, prosocial attitude, innovation attitude, and entrepreneurial self-efficacy. Ref. [91] argue that there need to be more studies that examine the relationship between innovation and profit in social entrepreneurship. Profit motivation may vary for social entrepreneurs.

Researchers [169] studied Italian entrepreneurs and found that less than half of the entrepreneurs surveyed chose entrepreneurship for higher financial income. Instead, non-monetary factors such as independence are more significant. A subsequent study [170] yielded similar results. Ref. [171] conducted a study on Canadian entrepreneurs and found that the primary goal of entrepreneurship was not to earn money but to increase personal development and adaptability.

Social entrepreneurship aims to improve the well-being of society through various means, such as prosocial motivation, profit motivation, and innovation [30,56,172–174]. The authors of [91] argue that prosocial entrepreneurs strive to enhance their well-being by

working towards the betterment of society, referred to as the "prosocial attitude" [30,82,93,175–181]. This attitude encompasses traits such as altruism, empathy, moral judgment, concern, compassion, guilt, self-glorification, self-esteem, status, and self-satisfaction.

Social entrepreneurs who prioritize the well-being of others may prioritize financial gain less, but they do expect greater social benefits [182]. Profit is a secondary motivation for them. They aim to generate profit while creating community value and promoting sustainability [77,183–185].

Research in social entrepreneurship has labeled social entrepreneurs "hybrid" entrepreneurs [186–189]. They were split into two categories by [190]: non-profit and hybrid (with profit and social goals) entrepreneurs. These hybrid entrepreneurs do expect to make a profit from their ventures. Nevertheless, this expectation may vary due to personal factors such as loan repayment, dependent children, family life, real estate projects, and health issues. However, some may expect little financial gain due to strong humanistic characteristics, a comfortable financial situation, or a non-materialistic lifestyle.

Social entrepreneurs can, but do not have to, engage in sustainable development [180]. For them, entrepreneurship is a way to improve social wealth by creating new firms or taking over existing ones to enhance the lives of current and future generations.

To summarize this section, the conventional perception of entrepreneurship as being solely focused on achieving financial gain is not accurate. Non-financial goals such as autonomy, personal ideas, job satisfaction, and responsibility also play a significant role in understanding the complexity of entrepreneurship. Research on social entrepreneurship has primarily focused on understanding the relationship between social entrepreneurship, prosocial attitude, innovation attitude, and entrepreneurial self-efficacy. However, there need to be more studies that examine the relationship between innovation and profit in social entrepreneurship. Profit motivation may vary for social entrepreneurs, and they may prioritize larger social benefits over financial gain. Social entrepreneurship aims to improve the well-being of society through various means, such as prosocial motivation, profit motivation, and innovation. Social entrepreneurs are labeled "hybrid" and may expect to make a profit; however, this expectation may vary due to personal factors.

*4.4. Research Gaps*

Although there has been extensive research conducted on entrepreneurship and profit, several gaps in the literature still need to be addressed. These gaps offer a roadmap for future research to better understand sustainable entrepreneurship's complexities. One such gap is related to the measurement of profit in sustainable entrepreneurship. While profit is not the sole objective for sustainable entrepreneurs, it is still an essential aspect of their businesses. Therefore, further research is needed to explore how sustainable entrepreneurs measure their profits and how these differ from traditional profit measurements. Additionally, it is necessary to investigate how sustainable entrepreneurs balance their financial, social, and environmental goals.

Another research gap relates to the motivations of sustainable entrepreneurs. While several studies have investigated the causes of social entrepreneurs, more research needs to be carried out to understand the motivations of sustainable entrepreneurs. Specifically, it would be beneficial to understand whether sustainable entrepreneurs are more motivated by financial rewards or by the desire to create a positive impact on society and the environment.

Moreover, there is a need to explore the role of sustainable entrepreneurship in promoting economic growth. Although sustainable entrepreneurship offers opportunities for entrepreneurs to generate profit through environmentally friendly business models, how this type of entrepreneurship contributes to economic growth is still being determined. Further research is needed to explore the potential economic benefits of sustainable entrepreneurship and how policymakers can promote this type of entrepreneurship to enhance economic growth.

Finally, there is a need to investigate how sustainable entrepreneurship can be scaled up to significantly impact the economy and society. While many sustainable entrepreneurs operate on a small scale, exploring how sustainable entrepreneurship can be scaled up to impact the environment and society significantly is necessary. This research could investigate the barriers and enablers of scaling up sustainable entrepreneurship and how policymakers can support the scaling up of sustainable entrepreneurship. By addressing these gaps, future research can better understand the nuances of sustainable entrepreneurship and provide valuable insights for practitioners and policymakers.

## 5. Conclusions

### 5.1. Research Implications

In recent years, sustainable entrepreneurship has become a pressing issue [18,19,23]. As the world population grows and the demands for natural resources increase, we must promote more sustainable businesses that positively impact the world. Sustainable entrepreneurship has many definitions [3,21]. For our purposes, we define it as a business that meets the needs of the present without compromising the ability of future generations to meet their own needs. In other words, sustainable entrepreneurship is about creating environmentally friendly businesses that use resources efficiently and have a positive social impact.

There are many reasons why sustainable entrepreneurship is essential [27,191]. Sustainable entrepreneurship is critical to the survival of our planet. Sustainable entrepreneurship is good for the economy. Sustainable companies are often more efficient and use resources more effectively, which can lead to cost savings. In addition, sustainable businesses often have a positive social impact, which can create jobs and support local economies. Sustainable entrepreneurship is good for society.

Sustainable businesses have a positive social impact that can help improve people's lives worldwide. There are many ways to create a sustainable business. Some companies may focus on using renewable energy, while others may work to reduce their environmental impact. Whatever the approach, the goal is to create an environmentally friendly business that uses resources efficiently and has a positive social impact.

### 5.2. Managerial and Practical Implications

Our article is truly unique as it presents a novel conceptual framework on sustainable entrepreneurship structured on the triple bottom line, which has not been used in entrepreneurship before. This approach has significant implications for individuals, organizations, and policymakers interested in sustainable entrepreneurship and its potential to drive economic growth while preserving the environment. This article offers insights into how individuals and organizations can create and implement policies and practices that promote sustainable entrepreneurship by providing an overview of sustainable entrepreneurship and its intersection with fields such as global economic growth, human resources management, and sustainability.

This article provides managerial and practical implications for sustainable entrepreneurship from various perspectives. Individuals can learn how to establish socially and environmentally responsible business models that create value for all stakeholders. Organizations can develop policies and practices that encourage employee participation in sustainable entrepreneurship, leading to a culture of sustainability and increased engagement. At a global level, sustainable entrepreneurship requires a joint effort between individuals, organizations, and policymakers to create a sustainable future. Additionally, the article highlights the importance of social entrepreneurship and how non-financial goals can play a significant role in the entrepreneurial journey. Overall, the report emphasizes the need for a more holistic approach to entrepreneurship that considers financial and social impact.

The themes of entrepreneurship and people, entrepreneurship and planet, and entrepreneurship and profit are interconnected, and they all have significant implications for CSR. Entrepreneurship has evolved beyond the conventional perception of profit-seeking to encompass social and environmental responsibility and financial gain. From an individual

perspective, entrepreneurship is about creating a socially and environmentally responsible business model. At an organizational level, it involves creating and executing policies and practices that support and promote sustainable entrepreneurship, encouraging employee participation, and establishing positive relationships with stakeholders. At a global level, it requires a joint effort between individuals and organizations to create a sustainable future that benefits people and the planet. Entrepreneurship and CSR go hand in hand, as entrepreneurs are uniquely positioned to integrate social and environmental considerations into their business models. By incorporating sustainable practices, entrepreneurs can create positive social and environmental impacts while generating profits. This integration of social and ecological responsibility with profit-seeking motivations characterizes CSR and social entrepreneurship. Ultimately, by prioritizing the well-being of people and the planet, entrepreneurs can drive positive social change and sustainable economic growth.

In a nutshell, this article provides a comprehensive overview of sustainable entrepreneurship, its intersection with various fields, and its potential to drive economic growth while preserving the environment. By highlighting the importance of creating socially and environmentally responsible business models, policies, and practices, this article offers insights into how individuals, organizations, and policymakers can contribute to a sustainable future.

### 5.3. Contributions to Theory from Various Perspectives

One notable contribution of this study is integrating Elkington's triple bottom line framework into the analysis of sustainable entrepreneurship. This framework, encompassing the dimensions of people, planet, and profit, provides a comprehensive lens for understanding the multifaceted nature of sustainability and its intersections with entrepreneurship. By adopting this framework, our article expands the theoretical underpinnings of sustainable entrepreneurship beyond conventional perspectives.

Our research sheds light on the intersection of sustainable entrepreneurship with diverse fields, including global economic growth, human resources management, and sustainability. By exploring these connections, we contribute to the theoretical discourse on how sustainable entrepreneurship can drive economic growth while preserving the environment. Moreover, we provide insights into how individuals and organizations can develop policies and practices that promote sustainability within the context of entrepreneurship.

Another significant contribution of this study lies in its emphasis on social entrepreneurship and recognizing non-financial goals within the entrepreneurial journey. By highlighting the importance of social and ecological responsibility alongside profit-seeking, our article underscores the role of sustainable entrepreneurship in driving positive social change and sustainable economic growth. This perspective enriches the theoretical landscape by advocating for a more holistic approach considering the financial and social impact.

Building upon the established triple bottom line framework, our study acknowledges the growing importance of governance as a fourth pillar in sustainable entrepreneurship. Good governance balances environmental, social, and economic considerations, facilitating long-term sustainability. By recognizing the significance of governance, we contribute to the theoretical discourse on the effective management and decision-making processes required to achieve sustainable outcomes.

### 5.4. Implications for Different Researchers

Sustainable entrepreneurship is a vital issue in the modern world. However, researchers from different countries, work cultures, business philosophies, political settings, and economic systems may have different perspectives. It has been defined as a business that meets the needs of the present without compromising the ability of future generations to meet their own needs. Sustainable businesses are often more efficient and use resources more effectively, leading to cost savings and a positive social impact. Researchers can use the triple bottom line framework to examine sustainable entrepreneurship, which has

not been used in entrepreneurship before. The article provides managerial and practical implications for sustainable entrepreneurship from various perspectives.

Sustainable entrepreneurship requires a joint effort between individuals, organizations, and policymakers to create a sustainable future. The themes of entrepreneurship and people, entrepreneurship and planet, and entrepreneurship and profit are interconnected and have significant implications for CSR. Entrepreneurs are uniquely positioned to integrate social and environmental considerations into their business models, creating positive social and environmental impacts while generating profits. By prioritizing the well-being of people and the planet, entrepreneurs can drive positive social change and sustainable economic growth. This article provides managerial and practical implications for sustainable entrepreneurship, which can benefit individuals, organizations, and policymakers interested in creating a sustainable future. By prioritizing social and ecological responsibility, entrepreneurs can drive positive social change while generating profits.

### 5.5. Future Research Directions

Elkington's triple bottom line approach [192] has been widely accepted as the gold standard for corporate sustainability [6,13,18,20,22,25,193]. Nevertheless, the literature suggests that a fourth pillar may be added to the framework-governance. Good governance is essential for any organization but crucial for those committed to sustainability. Why? Because sustainability concerns strike a balance between environmental, social, and economic problems.

Furthermore, good governance remains the only way to maintain this balance in the long term. Several reasons explain why management is so crucial for sustainability [16,91,194]. Below, we have listed the main reasons:

- First, it ensures that all stakeholders have a say in decision making. This feature is essential because sustainability concerns not just environmental issues but also social and economic issues. If one group of stakeholders stays out of the decision-making process, the whole system is at risk of becoming unbalanced.
- Second, good governance helps make all decisions in the best interests of all stakeholders. This point is vital because sustainability is about creating value for all stakeholders-not just shareholders. If decisions benefit one group of stakeholders at the expense of another, then the system is not sustainable.
- Third, good governance helps to ensure that decision-makers are accountable for their actions. Accountability is necessary because sustainability ensures that our actions today do not jeopardize the ability of future generations to meet their own needs. If decision-makers are not held accountable for their actions, they will be more likely to make short-sighted decisions that could have long-term negative consequences.
- Fourth, good governance ensures that organizations are transparent in their decision-making. Transparency is crucial because sustainability creates trust between organizations and their stakeholders. Organizations must be more transparent in their decision making so that stakeholders will trust them and the system will be sustainable [195,196].

To measure the impact of good governance on sustainability, several measurement tools can be employed. These include stakeholder analysis, sustainability reporting framework, governance assessment tool, and social responsibility index. Stakeholder analysis involves identifying all stakeholders involved in a given sustainability project or program, understanding their interests and power, and assessing their level of involvement in the decision-making process. Sustainability reporting framework involves measuring the sustainability performance of an organization against established sustainability indicators. Governance assessment tool involves measuring the effectiveness of decision-making processes, identifying areas of weakness, and suggesting potential improvements. Social responsibility index involves evaluating an organization's performance in terms of social and environmental responsibility, transparency, and accountability.

*5.6. Limitations*

As with all studies, we acknowledge the limitations of our research. First, one may require more than our study's inclusion and exclusion criteria. Indeed, a literature review can always be subject to criticism [126,167]. Although our review article is comprehensive, we cannot guarantee that we include all relevant work regarding sustainable entrepreneurship. Nevertheless, we assume that the literature covered represents all the results found in all the literature, and our literature review is as comprehensive as possible. We also aim to cover contemporary sustainable entrepreneurship's foundations and the most important topics.

In addition, objectivity concerns may appear, resulting from the data analysis. It is true that literature reviews and data collection analyses require interpretation and are subjective. Other researchers could identify several core themes and interpret the results differently. To counteract this limitation, we hope that the number of studies reviewed helps reduce the bias.

Furthermore, our framework could be improved by being more systematic. Although we discussed several interrelationships under each component, this discussion may still need to continue, and researchers can identify additional connections. That will be the work of further research. Nevertheless, the representation framework offers a helpful building block with a potential advance in our current understanding of sustainable entrepreneurship.

To reiterate, we acknowledge that our study has limitations and suggest that future research could address these limitations by using different inclusion and exclusion criteria, reducing bias, and developing more systematic frameworks. These criteria allow researchers to expand the scope of their study and include a more comprehensive range of relevant work regarding sustainable entrepreneurship. Additionally, the data analysis in this study is subjective. It may be subject to interpretation, and future research could aim to reduce bias by reviewing a more significant number of studies. Finally, our framework could be improved by being more systematic, considering all the relevant factors and interrelationships between different components of sustainable entrepreneurship. This would help to expand our understanding of sustainable entrepreneurship and provide more robust evidence to guide policy and practice in this field.

**Author Contributions:** Conceptualization, R.Y.C., A.C., S.H., I.P. and S.D.; methodology, R.Y.C., A.C., S.H., I.P. and S.D.; software, R.Y.C., A.C., S.H., I.P. and S.D.; validation, R.Y.C., A.C., S.H., I.P. and S.D.; investigation, R.Y.C., A.C., S.H., I.P. and S.D.; resources, R.Y.C., A.C., S.H., I.P. and S.D.; writing—original draft preparation, R.Y.C., A.C., S.H., I.P. and S.D.; writing—review and editing, R.Y.C., A.C., S.H., I.P. and S.D.; visualization, R.Y.C., A.C., S.H., I.P. and S.D.; supervision, R.Y.C., A.C., S.H., I.P. and S.D.; project administration, R.Y.C., A.C., S.H., I.P. and S.D. All authors have read and agreed to the published version of the manuscript.

**Funding:** This research received no external funding.

**Institutional Review Board Statement:** Not applicable.

**Informed Consent Statement:** Not applicable.

**Data Availability Statement:** Not applicable.

**Acknowledgments:** In this article, the authors searched the literature review, wrote and articulated the ideas, and integrated the references. They wrote the first draft of the article's text on this basis. Then, they used DeepL, Chat GPT3, and Grammarly tools to improve the writing. All the authors contributed equally to the article. They all designed, wrote, and revised the article.

**Conflicts of Interest:** The authors declare no conflict of interest.

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
