# Peer review of "Corporate Social Responsibility and Entrepreneurial Ventures: A Conceptual Framework and Research Agenda"

_sustainability, doi:10.3390/su15118849_

Round 1
Reviewer 1 Report
Thank you for the opportunity to revise this paper. I think the paper is generally fine but I have a few suggestions for the authors.
First of all, the definitions provided in the beginning of the introduction, are they literal quotations? If so they should be in quotation marks and the page should be mentioned. If not, it would be convenient to mention various references from which the authors create their own definition (for the second one).
The introduction presents well what the literature has analyzed regarding the main concepts, but I do not see clearly stated the gap and the research question(s) the study aims to fill and address.
You indicate in the last paragraph of the intro that you provide a survey, but in reality your paper is a conceptual one (surveys are usually associated to empirical research). I would recommend that you describe more clearly and without ambiguities the specific contribution of your paper.
I would recommend you to check the format, as sometimes the fonts are different, and even the colors!
In p. 11 there are two paragraphs that start the same (At the organizational level,….), I recommend changing the writing.
Also please note that sometimes you indicate the number of the reference, and other times you indicate the name of the author and the number (and not only when the author is followed of words like “argue” or similar, see for instance section 4.2). Besides, sometimes there are extra spaces and sometimes missing spaces when including a number of a reference.
Good luck!
Reviewer 2 Report
· The title of the article is not attractive and catchy. Use more powerful and meaningful words in the title to get the readers to proceed further. Similarly, the abstract cannot convince the readers of the content of this article. Please follow the standard abstract writing format for conceptual papers or meta-analyses.
· The authors attempt to use a systematic review to gather all available empirical and conceptual-based research outcomes. But not able to engage the assessment using appropriate themes concerning entrepreneurship and CSR. Lacks systematic methods to obtain responses to a specific question.
· The writing is not engaging. It was merely illustrating what others have written. I would expect the synthesis to be presented in matrix tables to showcase the strength of the authors in the review and advancement of new knowledge, guide to practice and further research.
· I strongly encourage the authors to reformat the writing and create viable themes within entrepreneurship and CSR rather than writing a piece of knowing facts. The paper is just a literature review with definitions and characteristics of entrepreneurship and CSR. Surprisingly, the end includes conclusion remarks discussing sustainable entrepreneurship, which is not the earlier parts of the write-ups.
· The paper should emphasise why there is a need for entrepreneurship and CSR, although many other issues could link with entrepreneurship. What’s new? What is the use of writing the know information? How could researchers benefit from the input from this article/argument? What’s best in the future following the discussions – positivism or phenomenological approach? Besides research, how could the layperson benefit from this type of article?
There are no significant issues with the use of the English language. Still, the article is not in a critical view, has poor synthesis, poor usage of systemic terminologies, and many parts of the sentences disengage.
Reviewer 3 Report
Page 1. I recommend the authors avoid ‘informal’ language, such as “Now that we know the definitions, we can start…” (p. 1). Please, use a more scientific writing style, as you intend to publish in a Scopus Q1 journal.
Page 3. CRS must be CSR
Page 3. References are wrongly cited in the main text of the paper, as you mix Harvard and APA, as with Petrella et al. [27]…, Austin et al. [28],… Dees [29], Dees and Anderson [30], Harrison, et al. [31], Klein and Harrison [32], and so on. I suggest the authors use Harvard only.
Page 4. Please, avoid trivial issues in the paper, such as “Entrepreneurship driven by necessity is often referred to as ‘necessity entrepreneurship.’” It is obvious! Please, rewrite it.
Page 4. Include data in “Although a positive correlation exists between business creation, ownership, and economic growth (as highlighted in the Global Entrepreneurship Monitor Report)” to show how strong is this positive correlation.
Page 10. I suggest the authors include specific references related to the two main streams related to the literature on sustainable entrepreneurship.
Page 15. The Discussion section is too short and poor.
I find a strong disequilibrium in this paper between the state of the art and the discussion and conclusions section, with no empirical analyses having been made. As a result, conclusions are not firmly supported, and no scientific advances can be observed in the paper related to the topic analyzed. In my opinion, using Artificial Intelligence (Chat GPT3) does not strengthen the quality of the arguments and literature review used in the paper. That is why many ideas in the article are vague and not strongly grounded.
In my opinion,
Reviewer 4 Report
Improvement required.

Round 2
Reviewer 1 Report
I have no further comments

Author Response
Please see the document.

Reviewer 2 Report
1. Entrepreneurship and corporate social responsibility (CSR) are extensively studied conceptually, and even empirical research is available. How is this paper unique and could contribute significantly to the business venture/entrepreneurship field?
2. The discussion of the work of various scholars embedded in the paper is not critical. It is merely statements of scholars illustrated before. I strongly recommend that the authors debate the CSR issues within the frame of the business venture. Develop a systematic literature review and write the gist in well-developed themes/perspectives, which will contribute better.
3. Presenting the summary in a more comprehensive format using tables will show the credibility of the authors in synthesising the content and attract potential readers. I saw a few tables in the paper but not in a quality synthesis.
4. The proposed research gaps are also simple. How future research benefits from the proposed gaps that have been addressed and empirically tested.
5. How researchers from different countries, work cultures, business philosophies, political settings and economic systems use the findings of this conceptual review paper and further undertake empirical research that contributes to the body of knowledge.
6. Bring into discussion several theories supporting CSR and entrepreneurship bonding. Suggest measurement tools based on the review.
7. A matrix table with finding from different nations and business disciplines will be attractive.
8. The conclusion section is not robust, repeating the content from the body.
9. I appreciate the authors' efforts in writing this paper; the way the content is presented not attractive. Improvise the flow of the content using subsections with extensive supporting sources.
10. The limitation stated in this paper should have been considered, researched and included as the findings in this paper.
Acceptable
Author Response
Please see the document.

Reviewer 3 Report
Firstly, congrats on your efforts to increase the quality of the previous version of your paper.
If possible, I suggest the authors slightly change some of their sentences to avoid beginning their paragraphs with [number], as it is uncommon to write this way. For example, the authors have written,
[96] posits that entrepreneurs and social entrepreneurs share many similarities, including ambition, leadership, problem-solving capabilities, disruptive thinking, and persuasive communication skills.
[95] asserts that a project’s success in entrepreneurship is measured by its implementation and development, which involves transforming an idea into desired outcomes
[96] stresses that social entrepreneurship can promote the growth of social capital by encouraging a more inclusive and diverse community, including those with physical or social disabilities.
And so on.
If possible, I suggest transforming these sentences to insert the reference at the end of the sentence as follows,
Entrepreneurs and social entrepreneurs share many similarities, including ambition, leadership, problem-solving capabilities, disruptive thinking, and persuasive communication skills [96].
A project’s success in entrepreneurship is measured by its implementation and development, which involves transforming an idea into desired outcomes [95].
Social entrepreneurship can promote the growth of social capital by encouraging a more inclusive and diverse community, including those with physical or social disabilities [96].
And so on
Page 20. I suggest the authors rename “5. Discussion and conclusion” as “5. Conclusion”, as you have discussed the different aspects of entrepreneurship as a topic in the previous pages of your paper.
The font style used in Tables 2 and 3 seems to differ from those used in the main text and Table 1. Please, check it.
Author Response
Please see the document.

Round 3
Reviewer 2 Report
Good improvement. Please include contributions to theory from various perspectives.
Author Response
Please see the document.
